# Association of Periodontal Status, Number of Teeth, and Obesity: A Cross-Sectional Study in Japan

**DOI:** 10.3390/jcm10020208

**Published:** 2021-01-08

**Authors:** Norio Aoyama, Toshiya Fujii, Sayuri Kida, Ichirota Nozawa, Kentaro Taniguchi, Motoki Fujiwara, Taizo Iwane, Katsushi Tamaki, Masato Minabe

**Affiliations:** 1Division of Periodontology, Department of Oral Interdisciplinary Medicine, Graduate School of Dentistry, Kanagawa Dental University, 82 Inaoka-cho, Yokosuka, Kanagawa 238-8580, Japan; t.fujii@kdu.ac.jp (T.F.); kdsyr61@gmail.com (S.K.); k.taniguchi@kdu.ac.jp (K.T.); minabe@kdu.ac.jp (M.M.); 2Division of Prosthodontic Dentistry for Function of TMJ and Occlusion, Department of Oral Interdisciplinary Medicine, Graduate School of Dentistry, Kanagawa Dental University, 82 Inaoka-cho, Yokosuka, Kanagawa 238-8580, Japan; nozawa@kdu.ac.jp (I.N.); fuji12422003@yahoo.co.jp (M.F.); tamaki@kdu.ac.jp (K.T.); 3School of Nutrition and Dietetics, Faculty of Health and Social Services, Kanagawa University of Human Services, 1-10-1 Heisei-cho, Yokosuka, Kanagawa 238-8522, Japan; iwane-d2k@kuhs.ac.jp

**Keywords:** chewing ability, infection, inflammation, periodontal medicine

## Abstract

Recent reports have shown an association between obesity and periodontitis, but the precise relationship between these conditions has yet to be clarified. The purpose of this study was to compare the status of periodontitis, tooth loss, and obesity. Participants comprised 235 patients at the Center for Medical and Dental Collaboration in Kanagawa Dental University Hospital between 2018 and 2020. Clinical examinations such as blood testing, body composition analysis, periodontal measurement, assessment of chewing ability, salivary testing, and oral malodor analysis were performed. Periodontal inflamed surface area (PISA) was significantly associated with the number of teeth and body mass index (BMI). The number of teeth was negatively associated with age, but positively with chewing ability. Chewing ability was associated negatively with age, and positively with high-sensitivity C-reactive protein (hsCRP). The level of methyl-mercaptan in breath and protein and leukocyte scores from salivary testing were positively associated with PISA. The rate of insufficient chewing ability was increased in subjects with hemoglobin (Hb)A1c ≥ 7%. The high PISA group showed increased hsCRP. BMI as an obesity marker was positively associated with PISA, indicating periodontal inflammation. Chewing ability was related to serum markers such as HbA1c and hsCRP.

## 1. Introduction

Obesity is becoming a serious problem for public health and is associated with many kinds of health concerns, such as diabetes mellitus, hypertension, dyslipidemia, and cardiovascular diseases [1,2]. Periodontitis is a common chronic inflammatory disease, characterized by gingival inflammation and destruction of tissue around the teeth, resulting in tooth loss [3]. Recent reports showed a strong association between periodontitis and general health. Periodontitis and tooth loss may be risk factors for cardiovascular disease [4]. Ischemic heart disease remains the single largest cause of death in countries of all income groups with a rise in obesity and diabetes mellitus [5]. Patients with periodontitis should be advised that there is a higher risk for cardiovascular diseases, such as myocardial infarction or stroke [6].

An association between obesity and periodontitis has recently been recognized [7]. The available evidence suggests a significantly positive association between periodontal disease and obesity not only in adults but also in children [8]. Obesity may influence host susceptibility to periodontopathic bacteria via inflammatory reactions [9]. Specific periodontal pathogens may thus play roles in obese patients [10]. Although the underlying pathophysiological mechanism remains unclear, the development of insulin resistance as a consequence of a chronic inflammatory state and oxidative stress could be implicated in the association between obesity and periodontitis [11]. A recent report showed that obesity is also associated with a higher risk of tooth loss [12]. Periodontal diseases and chewing disorders are associated with poor nutrition [13]. Periodontitis patients exhibited a higher body mass index (BMI) and altered diet practices [14]. A change in food intake might be a factor connecting oral health and obesity.

The world is aging, and Japan is the most aging country in the world [15]. Although the percentage of overweight/obese people in Japan is lower than that in other countries [16], many Japanese patients suffer from diabetes and periodontitis. The associations among periodontal disease, tooth loss, and obesity have been reported; however, data on the Japanese population is limited. The aim of the present cross-sectional study was thus to clarify relationships between periodontitis, tooth loss, and obesity within the Japanese population.

## 2. Materials and Methods

### 2.1. Study Population

Subjects were recruited from patients at the Center for Medical and Dental Collaboration in Kanagawa Dental University Hospital between 2018 and 2020. Data were obtained from 235 subjects in this study. Inclusion criteria were as follows: age ≥ 20 years; and consent to participate in this study. The criterion of 20 years old was used according to the definition of adult in Japan. Exclusion criteria were as follows: antibiotic intake within the past 2 months; severe systemic infection; or pregnancy or lactating status. The ethics committees of the School of Dentistry at Kanagawa Dental University approved the present study (approval no. 665), and the protocol conformed to the 1975 Declaration of Helsinki, as revised in 2013. Each participant was informed of the aims and procedures of the study. Written informed consent was provided by all participants prior to participation.

### 2.2. Clinical Examinations

All examinations were performed at the Center for Medical and Dental Collaboration in Kanagawa Dental University Hospital. General information on subjects such as age and sex were collected from medical records. Body composition analysis was performed using an analyzer (InBody 460; InBody Japan, Tokyo, Japan), and body indices such as height, weight, and BMI were recorded. Peripheral blood samples were obtained and serum levels of high-sensitivity C-reactive protein (hsCRP) and hemoglobin (Hb)A1c were measured.

Two trained periodontists (N.A. and M.M.) counted the number of residual teeth, excluding wisdom teeth. Probing pocket depth and bleeding on probing at six points per tooth for all teeth were measured using a manual probe (PCP-UNC 15; Hu-Friedy, Chicago, IL, USA). Using these periodontal parameters, periodontal inflamed surface area (PISA) was calculated as previously described [17].

Chewing ability was calculated using a Gluco Sensor GS-II (GC, Tokyo, Japan) in accordance with the instructions from the manufacturer. Oral malodor was objectively evaluated using OralChroma (FIS, Itami, Japan) as previously described [18]. The salivary test was performed using Sill-Ha (Arkray Inc., Kyoto, Japan) in accordance with the instructions from the manufacturer. Briefly, subjects rinsed their mouth with a dedicated solution for 10 s, then saliva samples were obtained with the solution. The oral rinse solution was applied to a test strip and placed in the instrument for testing. Protein score and leukocyte score were calculated from this salivary test, which indicated the level of inflammation in the gingiva.

### 2.3. Statistical Analysis

The Shapiro-Wilk test was performed to test the normality of data distributions. Numerical data are presented as mean ± standard deviation for parameters showing normal distributions and as median and interquartile range for skewed distributions. Spearman’s correlation coefficient was used to calculate correlations between values. The chi-square test was performed to compare the subject rate with insufficient chewing ability. The Wilcoxon test was used to compare differences in hsCRP between groups. JMP version 14.2.0 software (SAS Institute Inc., Cary, NC, USA) was used for all statistical analyses. Values of *p* < 0.05 were considered statistically significant.

## 3. Results

The characteristics of the subjects in this study are shown in Table 1. BMI ranges of 25–29.9 for overweight and ≥ 30 for obesity were used, because most of the epidemiologic data on obesity are based on this classification [19].

The relationships between PISA and parameters such as number of teeth, age, BMI, hsCRP, HbA1c, and chewing ability are shown in Figure 1.

PISA was significantly associated with the number of teeth and BMI. Associations between number of teeth and parameters such as age, BMI, hsCRP, HbA1c, and chewing ability are shown in Figure 2. The number of teeth was negatively associated with age, but positively with chewing ability.

Figure 3 shows the relationship between chewing ability and values like age, BMI, hsCRP, and HbA1c. Chewing ability was negatively associated with age, and positively with hsCRP.

Figure 4 shows the associations between BMI and values such as age, hsCRP, and HbA1c. BMI was associated with serum levels of hsCRP and HbA1c.

Figure 5 shows the associations between PISA and scores for halitosis and salivary test results. The level of methyl-mercaptan in breath was positively associated with PISA, as were protein score and leukocyte score from salivary testing.

Table 2 shows an association between chewing ability and HbA1c level. The rate of insufficient chewing ability (<150 mg/dL) was 22% in subjects with HbA1c < 7%, compared to 44% in subjects with HbA1c ≥ 7% (*p* = 0.047, chi-square test).

Figure 6 compares serum hsCRP levels between subjects with low and high PISA. A cut-off of 300 mm^2^ was used for PISA. The high-PISA group displayed increased levels of hsCRP (*p* = 0.046, Wilcoxon test).

## 4. Discussion

In this study, PISA was positively associated with BMI, while the number of teeth was not associated with BMI. Moreover, we found that chewing ability was related to systemic markers such as HbA1c and hsCRP. Insufficient chewing ability was found in those with HbA1c ≥ 7.

A relationship between periodontitis and obesity has been widely recognized [7,8,9,10,11], and the association was confirmed in the Japanese population [20]. The present study confirmed that periodontal destruction was increased in obese patients (Figure 1). Vallim et al. reported that obesity could represent a risk factor for tooth loss over 5 years [21], but no relationship between BMI and number of teeth was identified in our study (Figure 2). Other measures of obesity such as percent body fat and body fat volume might be associated with tooth loss.

Links between periodontal disease or tooth loss and diabetes are also known. Tooth loss and periodontal attachment were increased by hyperglycemia in individuals with diabetes [21]. In our data, however, the HbA1c level was not associated with PISA or the number of teeth (Figure 1 and Figure 2). The periodontal treatment reportedly improved HbA1c by reducing CRP [22]. An association between periodontal disease and hyperglycemia might be seen in specific patients with increased systemic inflammatory markers via periodontal inflammation. In our study, elevated levels of hsCRP were found in patients with increased PISA (Figure 6), while the hsCRP level was not associated with PISA (Figure 1). An association between glycemic control and systemic inflammation in people with established diabetes has been suggested [23]. Periodontitis may impair blood glucose regulation in healthy subjects in conjunction with elevated CRP levels [24]. More precise classifications might be effective in clarifying these relationships.

The present study also measured chewing ability. Masticatory dysfunction may be an important risk factor for mortality [25]. After controlling for possible confounding factors, the number of functional teeth and periodontal status were common factors associated with malocclusion [26]. The importance of mastication assessment in the diagnosis of periodontitis patients has been proposed [27]. As a result, chewing ability was related to systemic markers such as HbA1c and hsCRP (Figure 3, Table 2). To the best of our knowledge, this is the first report to investigate the relationship between chewing ability and serum glycemic and inflammatory markers. A lower number of teeth was associated with lower masticatory ability, which in turn was associated with lower plasma albumin levels and lower BMI [28]. A high level of HbA1c was suggested to be associated with poor masticatory function and severe periodontitis [29]. In the present study, chewing ability was associated with the number of teeth (Figure 2), but appeared unrelated to PISA (Figure 1) and BMI (Figure 3). Because chewing ability was related to oral health-related quality of life and general health [30], we should consider mastication in periodontal medicine.

Nutrition could be a crucial link between the oral condition and systemic health. Proper food intake is essential to maintain a healthy life. Obesity, diabetes, and other chronic non-communicable diseases (NCDs) are increasing globally, and malnutrition with inappropriate food intake leads to health issues [31]. The inclusion of nutrition is strongly recommended as a key focal point for all health professionals [32]. Chewing ability can influence food intake, and malnutrition induces insufficient general conditions. Nutrition might play a specific role in periodontal medicine. Further investigations regarding food intake are needed.

A strong association is well known to exist between periodontal and systemic diseases. A score reflecting the total inflammatory burden of periodontitis has been considered necessary, particularly in the field of periodontal medicine. Periodontal epithelial surface area (PESA) and PISA were thus proposed by Nesse et al. [17]. PESA indicates the surface area of the entire periodontal pocket epithelium, while PISA reflects the surface area of the bleeding pocket epithelium and the inflammatory burden posed by periodontitis. This conversion of periodontal inflammation to an individual value using PISA facilitates communication with healthcare workers outside of dental specialties. As a result, PISA has gained use in many studies in the field of periodontal medicine [33,34,35]. On this basis, we used PISA in this study to indicate periodontal inflammation.

Salivary testing was performed in this study and revealed that protein score and leukocyte score were positively associated with PISA. The utility of salivary testing has been confirmed in various reports [36,37,38]. Kim et al. [36] conducted a study of 10 male and female adults and indicated the measurement principles of salivary testing. Swiatkowska et al. [37] assessed 25 patients and showed that the parameters estimated by the analyzer correlated with the results from common salivary kits and oral health indices. Irie et al. [38] investigated 125 pediatric patients, between 3 and 18 years old. They found that leukocyte and protein scores changed according to gingival inflammation. Although those researchers investigated relatively small numbers of subjects, the utility of salivary analyzers was confirmed in the present study of 235 participants. Prediction of risks for periodontal disease using simple evaluation methods is an important issue, so the use of this salivary test is effective in screening for periodontal diseases.

Recently, a new classification scheme for periodontitis has been adopted, using a multidimensional staging and grading system [39]. Staging is dependent on disease severity as well as the complexity of disease management, while grading provides supplemental information about the biological features of the disease. However, this new system still shows some difficulties with classification and its authors have mentioned the need for the development of methodologies to accurately assess periodontal tissues. To better understand periodontal disease and its effects on systemic health, issues such as inflammation, infection, level of destruction and oral function should be considered [40,41]. Both inflammation and oral function are particularly important to evaluate the effects of periodontal disease on systemic health. The new classification also pays attention to the number of teeth and occlusal function, and PISA received attention as an inflammatory marker in the grade classification.

Some limitations of this study must be recognized and considered. We did not account for the medical conditions of participants in this study, because a wide variety of diseases and medications were involved. However, information on systemic diseases and medication is important when considering different patient backgrounds. Next, participants in this study were not limited to those on their first visit to the dental hospital. Therefore, the phases of dental treatment differed between subjects. These issues should be kept in mind when interpreting the data. Moreover, the sample size calculation was not performed before the research.

In conclusion, BMI as an obesity marker was positively associated with PISA, a marker of periodontal inflammation. Relationships between chewing ability and serum markers such as HbA1c and hsCRP were also found. Associations of periodontal status, number of teeth, and systemic condition should be considered in clinical settings.

## Figures and Tables

**Figure 1 jcm-10-00208-f001:**
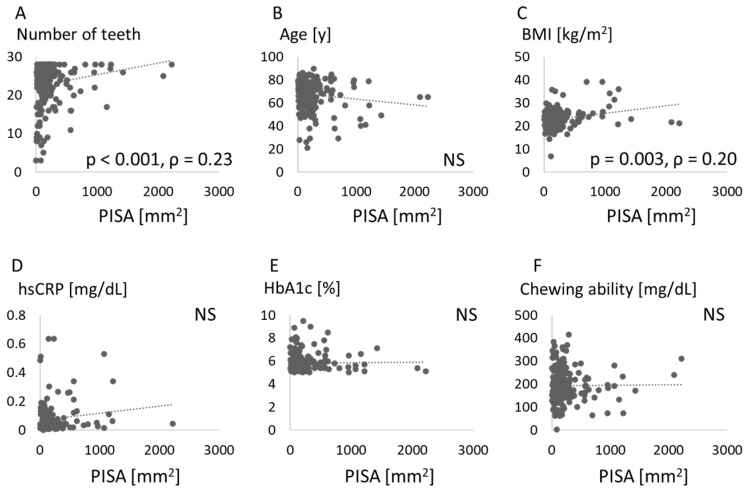
Association between each score and periodontal inflamed surface area. Number of teeth (**A**), age (**B**), body mass index (BMI) (**C**), high-sensitivity C-reactive protein (hsCRP) (**D**), hemoglobin (Hb)A1c (**E**), and chewing ability (**F**) are compared with periodontal inflamed surface area (PISA). NS: not significant.

**Figure 2 jcm-10-00208-f002:**
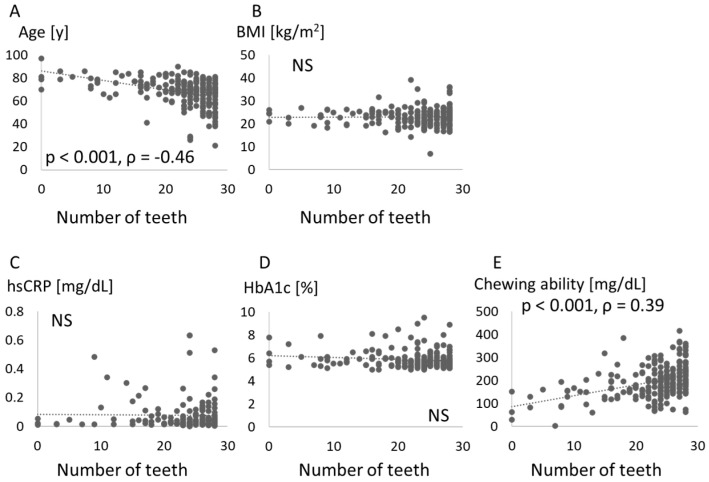
Association between each score and number of teeth. Age (**A**), body mass index (BMI) (**B**), high-sensitivity C-reactive protein (hsCRP) (**C**), hemoglobin (Hb)A1c (**D**), and chewing ability (**E**) are compared with number of teeth. NS: not significant.

**Figure 3 jcm-10-00208-f003:**
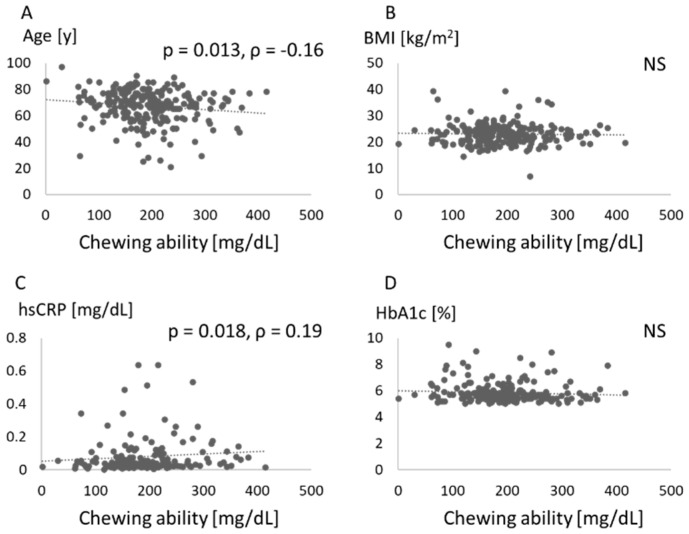
Association between each score and chewing ability. Age (**A**), body mass index (BMI) (**B**), high-sensitivity C-reactive protein (hsCRP) (**C**), and hemoglobin (Hb)A1c (**D**) are compared with number of teeth. NS: not significant.

**Figure 4 jcm-10-00208-f004:**
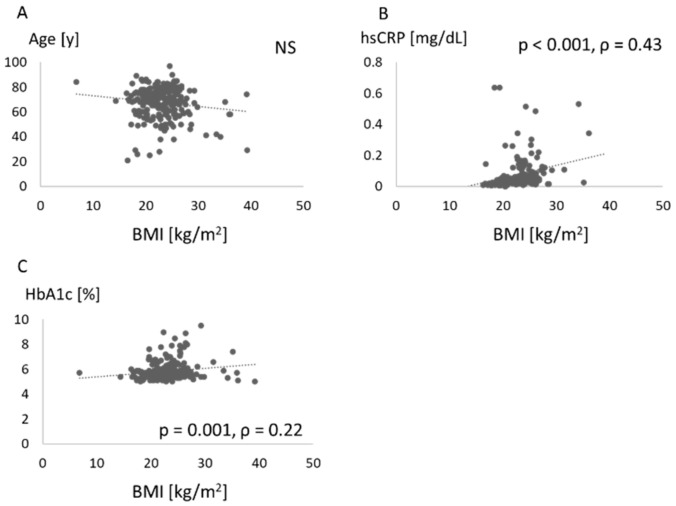
Association between each score and body mass index. Age (**A**), high-sensitivity C-reactive protein (hsCRP) (**B**), and hemoglobin (Hb)A1c (**C**) are compared with body mass index (BMI). NS: not significant.

**Figure 5 jcm-10-00208-f005:**
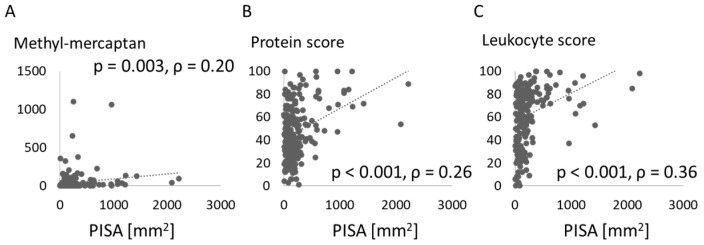
Association between oral parameters and periodontal inflamed surface area. Methyl-mercaptan in breath (**A**) and protein (**B**) and leukocyte (**C**) scores in saliva are compared with periodontal inflamed surface area (PISA).

**Figure 6 jcm-10-00208-f006:**
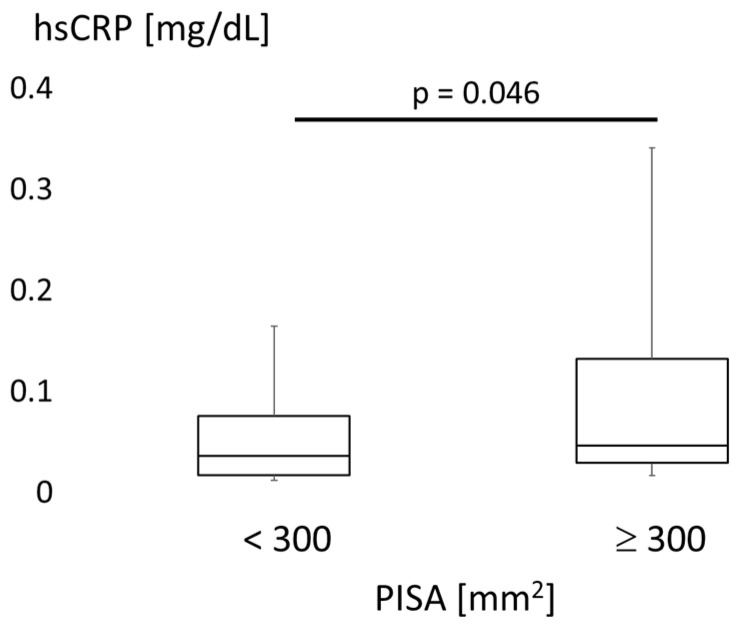
Comparison of high-sensitivity C-reactive protein. High-sensitivity C-reactive protein (hsCRP) is compared between high (≥300 mm^2^) and low (<300 mm^2^) periodontal inflamed surface area (PISA) groups. Box plots show medians, 25th and 75th percentiles as boxes, and 10th and 90th percentiles as whiskers. The Wilcoxon test was used.

**Table 1 jcm-10-00208-t001:** Subject characteristics.

Variables	
N	235
Female (n, (%))	155 (66%)
Age (years)	67.2 ± 12.6 ^1^
Height (cm)	157.5 ± 8.6 ^1^
Weight (kg)	57.6 ± 11.7 ^1^
BMI (kg/m^2^)	23.1 ± 3.9 ^1^
Overweight (BMI 25–29.9 kg/m^2^) (n, (%))	55 (23%)
Obese (BMI ≥ 30 kg/m^2^) (n, (%))	8 (3.4%)
Number of teeth	25 (21, 27) ^2^
hsCRP (mg/dL)	0.078 ± 0.112 ^1^
HbA1c (%)	5.85 ± 0.75 ^1^
Chewing ability (mg/dL)	192.0 ± 69.5 ^1^
Methyl-mercaptan	9 (4, 28) ^2^
Protein score	61.3 ± 25.0 ^1^
Leukocyte score	47.1 ± 23.4 ^1^

^1^ Data are shown as mean ± standard deviation in age, height, weight, body mass index (BMI), high-sensitivity C-reactive protein (hsCRP), hemoglobin (Hb) A1c, chewing ability, protein score, and leukocyte score. ^2^ Number of teeth and methyl-mercaptan are shown as median (interquartile range).

**Table 2 jcm-10-00208-t002:** Chewing ability and HbA1c.

	HbA1c < 7%	HbA1c ≥ 7%	*p*
Insufficient chewing ability ^1^	22%	44%	0.047 ^2^

^1^ Cut-off point for chewing ability was set as 150 mg/dL. Hb, hemoglobin. ^2^ The chi-square test was used to compare groups.

## Data Availability

The data presented in this study are available on request from the corresponding author.

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
