# Peer review of "Association of Periodontal Status, Number of Teeth, and Obesity: A Cross-Sectional Study in Japan"

_jcm, 2021, doi:10.3390/jcm10020208_

Round 1

Reviewer 1 Report

Thank you for the opportunity to review this article. The work is interesting, but some aspects should be taken into account.

Comments and suggestions for Authors:

  • The text is unjustified

Introduction:

  • Authors should include a literature review in the introduction.
  • Authors should improve the introduction including the latest articles published for example in the MDPI platform or others, about other research in this field
  • Authors should better justify the study, highlighting clearly the gap in the current knowledge. What is the novelty of the study?

Materials and Methods

  • Please discuss power calculation and how the sample size is adequate.
  • Why authors include exactly 235 subjects? How many invitations did you send? Please write more about recruitment of the study group.
  • Where were the measurements taken?
  • Why was the inclusion criteria the age ≥20, not 18?
  • The title of the present study is “Association of periodontal status, number of teeth, and obesity”. There is poor information about the category of body weight. How did the authors classify obesity? Please add appropriate references.

Result:

  • Table 1 should contain the full characteristics of the studied group, including BMI.
  • The authors should provide, for example, in a table the results of all performed measurements, including blood parameters.
  • There is also no information on how many people were overweight/obese etc.
  • The appearance and readability of figures should be corrected.

Reviewer 2 Report

Dear authors, 

congratulations for your work. 

I think that you could add more information in the introduction about the importance of oral health with general health. 

I ave just one question: why didn't you asked for the number of toothbrushing per day? it could be interesting to know if the inflammation gum could become from a bad hygiene. 

Author Response

Dear Reviewer 2,

Thank you very much for your kind review. Following is our reply to your comments.

I think that you could add more information in the introduction about the importance of oral health with general health. 

Thank you very much for your advice. We added introduction regarding the importance of oral health with general health.

I ave just one question: why didn't you asked for the number of toothbrushing per day? it could be interesting to know if the inflammation gum could become from a bad hygiene. 

We agree that gum inflammation via bad oral hygiene is an important factor. We are now collecting such information like the number of toothbrushing per day and usage of adjunctive tools such as interdental brush. We will assess the relationship between oral hygiene and general health in the future research. Thank you very much for your valuable comment.

We hope that this revised manuscript is suitable for publication in the Journal of Clinical Medicine. We deeply appreciate your consideration. 

Round 2

Reviewer 1 Report

Thank you for the opportunity to review this resubmission.  Authors have done a nice job addressing reviewers' comments. Thank you. I am ok with acceptance.